

# Imprint of Southern Ocean eddies on chlorophyll

Ivy Frenger[1,2,3,*], Matthias Münnich[2], and Nicolas Gruber[2,4]

[1]GEOMAR Helmholtz Centre for Ocean Research Kiel, Kiel, 24105, Germany.
[2]ETH Zurich, Environmental Physics, Institute of Biogeochemistry and Pollutant Dynamics, Zurich, 8092, Switzerland.
[3]Princeton University, Program in Atmospheric and Oceanic Sciences, Princeton, 08544, USA.
[4]ETH Zurich, Center for Climate Systems Modeling, Zurich, 8092, Switzerland.

**Correspondence:** Ivy Frenger (ifrenger@geomar.de)

**Abstract.** Although mesoscale ocean eddies are ubiquitous in the Southern Ocean, their spatial and seasonal association with phytoplankton has to date not been quantified in detail. To this end, we identify over 100,000 eddies in the Southern Ocean and determine the associated phytoplankton biomass anomalies using satellite-based chlorophyll-a (Chl) as a proxy. The mean eddy associated Chl anomalies ($\delta$Chl) exceed $\pm 10\%$ over wide regions. The structure of these anomalies is largely zonal, with

cyclonic, thermocline lifting, eddies having positive anomalies in the subtropical waters north of the Antarctic Circumpolar Current (ACC) and negative anomalies along the ACC. The pattern is similar, but reversed for anticyclonic, thermocline deepening eddies. The seasonality of $\delta$Chl is weak in subtropical waters, but pronounced along the ACC, featuring a seasonal sign switch. The spatial structure and seasonality of $\delta$Chl can be explained largely by lateral advection, especially eddy-*stirring*. A prominent exception is the ACC region in winter, where $\delta$Chl is consistent with a modulation of phytoplankton light exposure

caused by an eddy-induced modification of the mixed layer depth. The clear impact of eddies on phytoplankton may implicate a downstream effect on Southern Ocean biogeochemical properties, such as mode water nutrient contents.

## 1   Introduction

Phytoplankton account for roughly half of global primary production (Field et al., 1998). They form the base of the oceanic food web (e.g., Pomeroy, 1974) and drive the ocean's biological pump, i.e., one of the Earth's largest biogeochemical cycles with

major implications for atmospheric $CO_2$ and climate (Sarmiento and Gruber, 2006; Falkowski, 2012). Yet, our understanding of the processes controlling their spatio-temporal variations is limited (McGillicuddy, 2016), particularly at the oceanic mesoscale, i.e., at scales of 10 to 100 km. It is well established that eddies, the most prevalent mesoscale features that also dominate the ocean's kinetic energy spectrum (Stammer, 1997; Chelton et al., 2011b), affect phytoplankton in a major manner (e.g., Uz and Yoder, 2004; Bernard et al., 2007; Ansorge et al., 2009; Lehahn et al., 2011; Peterson et al., 2011; Xiu et al.,

2011; Nel et al., 2001; Kahru et al., 2007; Godø et al., 2012). In fact, eddies are among the most important drivers for the spatio-temporal variance of phytoplankton biomass (e.g., Doney, 2003) as has been noted already from the analyses of some of the very first ocean color satellite images of chlorophyll (Chl), a proxy for phytoplankton biomass (Gower et al., 1980). Despite decades of work since this discovery, the mechanisms governing the interaction of phytoplankton with mesoscale eddies remain poorly understood, even though there is a broad consensus that different sets of mechanisms dominate in different



regions and at different times, and that the different polarity of the eddies tends to induce signals of opposite sign (Denman and Gargett, 1995; Lévy, 2008; McGillicuddy, 2016).

Lateral advection arising from *stirring* of eddies has been argued to be a major driver globally. The argument is based on the observed correlation of the magnitude of eddy-associated chlorophyll anomalies, $\delta$Chl, and the larger-scale Chl gradient

ambient to eddies (Doney, 2003; Uz and Yoder, 2004; Chelton et al., 2011a; O'Brien et al., 2013). Further, it has been suggested that advection of Chl by eddies via *trapping*, i.e., the enclosing and dragging along of waters, causes $\delta$Chl (Gaube et al., 2014), particularly in boundary current regions characterized by steep zonal Chl gradients. Numerous other potential mechanisms through which eddies affect phytoplankton have been identified (e.g., McGillicuddy et al., 2007; D'Ovidio et al., 2010; Siegel et al., 2011; Gaube et al., 2013, 2014; Dufois et al., 2016; Gruber et al., 2011), including modifications of mixed layer depth,

vertical mixing, thermocline lifting, and providing of spatial niches. These mechanisms modulate the phytoplankton's light exposure, their nutrient availability or their grazing pressure, i.e., they affect their net balance between growth and decay. Thus, in contrast to the physical mechanisms of *stirring* and *trapping* where phytoplankton is being advected merely passively, these mechanisms create eddy-associated phytoplankton biomass anomalies by altering their biogeochemical rates.

Here, we aim (i) to provide a reference estimate of the long-term mean chlorophyll anomalies associated with eddies in the

different regions of the Southern Ocean, distinguishing anticyclones and cyclones, and (ii) to discuss the mechanisms likely causing the observed imprint. The Southern Ocean is a region rich in eddies (e.g., Frenger et al., 2015) and important for setting the global distribution of biogeochemical tracers (Sarmiento et al., 2004). Previous studies used eddy kinetic energy as a proxy for eddy activity rather than sea level anomalies (SLA), which does not allow a distinction by polarity of eddies (Comiso et al., 1993; Doney, 2003), did not focus on the Southern Ocean (Chelton et al., 2011a; Gaube et al., 2014), or lacked

a discussion of the seasonality of the relationship.

Our approach is to identify individual eddies based on satellite estimates of SLA and combine those with satellite estimates of Chl (Chelton et al., 2011a; Gaube et al., 2014). We discuss possible mechanisms playing a role based on large-scale Chl gradients (Doney, 2003; Chelton et al., 2011a; Gaube et al., 2014) and the local shape of the average imprint of eddies on Chl (Chelton et al., 2011a; Gaube et al., 2014; Siegel et al., 2011).

## 2   Methods and data

We first introduce our analysis framework before describing the methods and data sources. This permits us to explain the approaches we use to assess the potential mechanisms explaining the $\delta$Chl associated with Southern Ocean eddies.

### 2.1   Analysis framework

Fundamentally, eddies can lead to local phytoplankton biomass anomalies through either advective processes, i.e., the spatial

reshaping of existing gradients, or through biogeochemical fluxes and transformations that lead to anomalous growth or losses of biomass. In the following, we present these potential mechanisms in more detail, and how we estimate their importance.





### 2.1.1  Causes of $\delta$Chl by advective processes

Eddies may cause $\delta$Chl as they laterally move waters, i.e., advect waters including their Chl characteristics. This mechanism may lead to $\delta$Chl if (i) a lateral Chl gradient is present that is sufficiently steep at the spatial scale of the eddy-induced advection (Gaube et al., 2014), and (ii) the time scale of advection matches the time scale of the growth and loss of Chl (Abraham, 1998).

The time scale of Chl is order of days to weeks, possibly months, with the lower boundary representing roughly the reactivity time scale of Chl governed largely by the growth rate of the phytoplankton, and the upper boundary the potential maintenance of Chl concentrations via recycling of nutrients within the mixed layer. Concerning the spatial scale of advection by eddies, we distinguish two effects, labeling them *stirring* and *trapping*.

With *stirring*, we refer to the distortion of a large-scale Chl gradient due to the rotation of an eddy, as illustrated in Figure

1a (left column, with black arrows indicating the eddy rotation and associated advection), see also Siegel et al. (2011). The turnover time scale associated with the rotation of eddies is order of days to a few weeks which matches the time scales of Chl. The spatial scale of *stirring* is given by the spatial extent of an eddy and is somewhat larger than the eddy core, as defined based on the Okubo-Weiss parameter (Frenger et al., 2015), i.e., several tens to several hundred kilometers.

Next to *stirring*, eddies may advect material properties due to their intrinsic lateral propagation (Figure 1a, right column).

We refer to the ability of eddies to transport fluid along their propagation pathway in their core as *trapping*. The time scale of *trapping* is given by the typical lifespan of Southern Ocean eddies which is weeks to months (Frenger et al., 2015), i.e. it may match the longer time scale of Chl. Propagation speeds are small (an order of magnitude smaller than rotational speeds) which implies that the majority of eddies tends to die before they can propagate far. The fraction of very long-lived eddies that propagate over distances exceeding a few hundred kilometers is small (Frenger et al., 2015).

A necessary condition for *trapping* to happen is that the eddies' swirl velocity is larger than their propagation speed (Flierl, 1981), a condition generally met for mid- to high-latitude eddies (Chelton et al., 2011b). Indeed, observations of eddies carrying the signature of their origin in their cores support the *trapping* effect (Bernard et al., 2007; Ansorge et al., 2009; Lehahn et al., 2011), and so does the modeling study by Early et al. (2011). Yet, likely only few eddies are truly efficient in their *trapping* (Beron-Vera et al., 2013; Haller, 2015). They tend to continuously exchange some fluid with their surroundings, i.e., their

trapping is limited by them being permeable. Nevertheless, we expect eddies to be able to drag along some entrained waters for some time, hence displacing these waters for some distance as they propagate. This may be sufficient to displace waters from e.g., the south to the north of an ACC front along an intense Chl gradient, leading to $\delta$Chl through (permeable) *trapping*.

### 2.1.2  Causes of $\delta$Chl by biogeochemical processes

Eddies affect the biogeochemical/physical properties that control the rates of biogeochemical processes in their interior through

many mechanisms. These include, e.g., the stimulation of phytoplankton growth through enhanced nutrient concentrations or increased average light levels, or the modification of predator-prey encounter rates, affecting the mortality of phytoplankton (Figure 1b). Even though these effects have been analyzed and discussed for decades (see review by McGillicuddy, 2016), their overall impact on productivity continues to be an issue of debate. The canonical vertical pumping of nutrients by thermocline





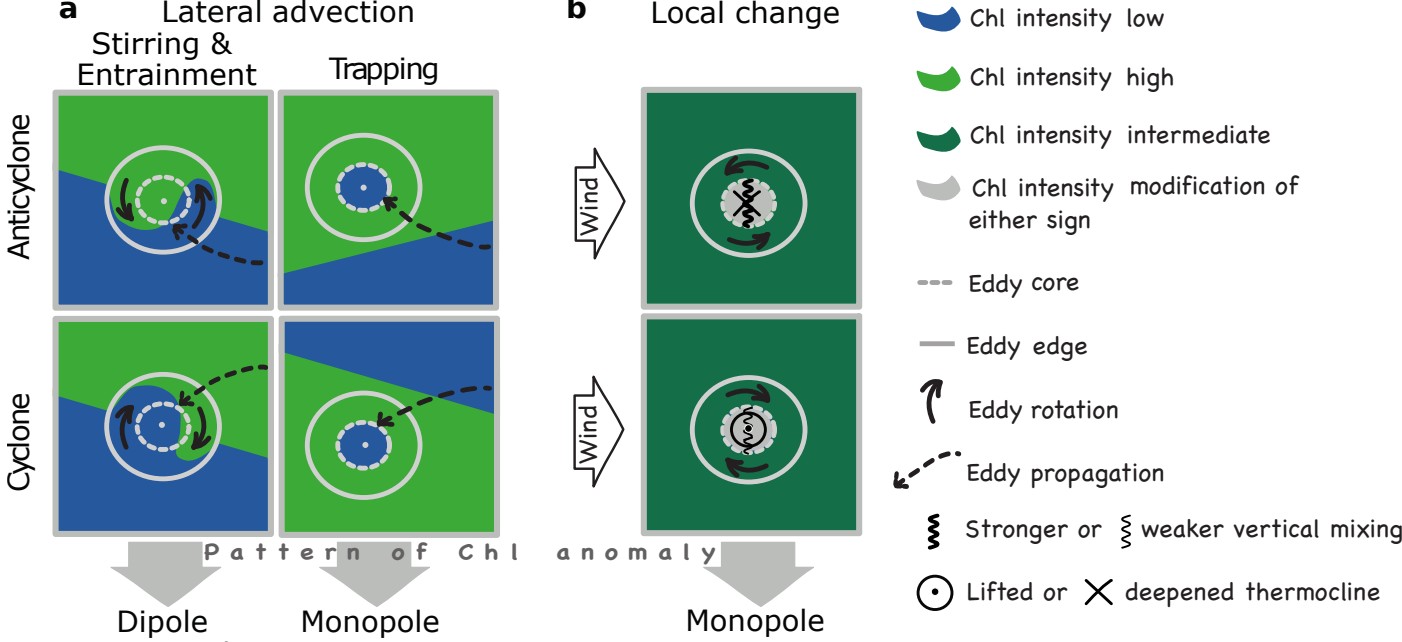

**Figure 1. Schematic illustrating the mechanisms of how eddies may impact chlorophyll (Chl)**, separated by anticyclones (top row) and cyclones (bottom row); **a** shows the effects of advection (lateral displacements) of Chl due to the eddies' rotational speed (*stirring*, left column) and lateral propagation (*trapping*, right column); *trapping* and *stirring* can cause $\delta$Chl of either sign, depending on environmental Chl gradients; **b** shows multiple potential effects eddies may have on Chl by affecting biogeochemical processes, including modification of nutrient supply and light exposure through thermocline lifting or deepening (circle/cross) and modified vertical mixing (wiggly lines), and eddy current-wind interactions (black and thick white arrows), in turn causing thermocline displacements; the local shape of $\delta$Chl is anticipated to look different depending on the mechanism active, i.e. a monopole $\delta$Chl is expected for all eddy-effects except for *stirring* where an asymmetric dipole is excepted (Figure inspired by Siegel et al. 2011, Figure 1).

lifting cyclones (Falkowski et al., 1991, indicated as black circle in Figure 1b) has been challenged to be a major player (Oschlies, 2002), and multiple other mechanisms have been proposed how eddies may affect phytoplankton biomass. These include a modification of vertical mixing through changes of stratification (wiggly lines in Figure 1b) and eddy current-wind interactions causing thermocline displacements (eddy swirl currents and winds are indicated as black and thick white arrows in

5   Figure 1b), resulting in modifications of nutrient supply and light exposure of phytoplankton (e.g., Llido, 2005; McGillicuddy et al., 2007; Mahadevan et al., 2008; Siegel et al., 2011; Xiu et al., 2011; Lehahn et al., 2011; Boyd et al., 2012; Mahadevan et al., 2012; Dufois et al., 2016). The prevailing lack of data of sufficiently highly temporally resolved sub-surface observations hampers a systematic large-scale observationally-based assessment of the role of effects of eddies on biogeochemical processes.




### 2.1.3 Assessing mechanisms causing $\delta$Chl

We employ two sets of approaches to assess the mechanisms causing $\delta$Chl. In the first set of approaches, we diagnose whether the environmental conditions are actually met for supporting a major contribution of a particular set of mechanism. Namely, we assess if lateral Chl gradients sufficiently support advective effects of eddies to explain $\delta$Chl.

In the second set of approaches we diagnose the spatial pattern associated with eddies as this spatial signature tends to differ between the two major sets of processes, i.e., advective processes and biogeochemical rates (Siegel et al., 2011). Eddies that are associated with, for instance, *stirring* are anticipated to have a dipole shape (Figure 1a, left column), as they distort the underlying gradient field, with the rotation of the eddy determining the orientation of the dipole. In contrast, most mechanisms associated with modifications of the biogeochemical rates cause a monopole pattern, irrespective of polarity (Figure 1b). This

is a consequence of the $\delta$Chl tending to be caused by anomalies in the nutrient supply or light availability, which are altered inside eddies in a radially symmetric manner. Also the *trapping* mechanisms tend to cause a monopole pattern of $\delta$Chl (Figure 1a, right column), but they can be distinguished from the rate-based mechanisms by either their history, or their tendency to trap the anomalies very tightly in the inner domain of the eddy. The rate based mechanisms, in contrast, often have monopole patterns that extend more broadly over the eddy, or are, in certain case, even strongest at the edges (McGillicuddy, 2016). Here,

we suggest effects on biogeochemical rates due to eddies to play a role in regions and seasons where the potential for advective effects is insufficient to explain the observed eddy-induced $\delta$Chl, i.e., we diagnose them largely as a residual.

Some complexity is added in the interpretation of the spatial pattern by the fact that the asymmetry of the dipole pattern arising from *stirring* causes also a monopole pattern (Figure 1a). Such an asymmetry was suggested by Chelton et al. (2011a) to arise from the westward propagation of eddies and the leading (mostly western) side of an eddy affecting upstream unperturbed

waters, resulting in a larger anomaly at the leading than the trailing side of an eddy, with the latter stirring already perturbed waters. Also, the eddy may entrain some of the westward upstream waters into its core, labeled here lateral entrainment or permeable *trapping* (Hausmann and Czaja, 2012; Frenger et al., 2015). Indeed, averaged over an eddy's core, *stirring* will only cause a net anomaly if the dipole associated with *stirring* is asymmetric. It is not obvious how to quantify such an asymmetry. Independent of its asymmetry, we will qualitatively discuss the potential maximum $\delta$Chl induced by *stirring*.

We note that advection by an ambient larger-scale flow does not affect the *stirring* mechanism. For instance, the Antarctic circumpolar flow in the Southern Ocean makes eddies propagate eastward in an Eulerian sense, nevertheless they propagate westward in a Lagrangian sense, relative to the ACC and ambient Chl.

### 2.2 Data

To assess the relationship between ocean eddies and Chl anomalies, we use the data set of Southern Ocean eddies and their

characteristics as derived and described in detail in Frenger et al. (2015). The data set contains about 1,000,000 snapshots of eddies identified in weekly maps of Aviso SLA and defined based on the Okubo-Weiss parameter. Eddies with positive and negative SLA are defined as anticyclones and cyclones, respectively. We consider here only eddies tracked over at least three weeks in the time period between 1997 and 2010, i.e., the operation time period of the SeaWIFS satellite-based sensor (see





below), in the region 30°S to 65°S.

For Chl we use the merged ESA GlobColour Project product (http://www.globcolour.info, case-1 waters, merged according to Maritorena and Siegel (2005) with a spatial and temporal resolution of 0.25° and one day, respectively. We choose a merged product for Chl as the merging doubles the spatial coverage of the daily data in the Southern Ocean, on average (Maritorena

et al., 2010). Of the data of the up to three available sensors, i.e., SeaWIFS (SeaStar), MODIS (Aqua) and MERIS (Envisat), SeaWIFS generally features the best spatio-temporal coverage, but its contribution drops below 40% in high latitudes and partly in the western ocean basins of the Southern Hemisphere. For these areas after 2002, SeaWIFS data were complemented with MODIS as well as MERIS data. We average the Chl data to weekly fields to match the temporal resolution of the eddy dataset.

To examine $\delta$Chl of eddies, we compare the Chl averaged over their core to background fields of Chl. For the latter, a monthly climatology of Chl proved not to be appropriate due to high spatio-temporal variability of Chl unrelated to eddies. Hence, we obtain the background fields the following way: we apply a moving spatio-temporal Gaussian filter (Weierstrass transform, spatial filter similar to e.g., Siegel et al. 2008, with 2$\sigma$ of 10 boxes/~200 km at 45°S, 8 boxes/~200 km and 1 week in longitudinal, latitudinal and temporal dimensions, respectively) to each of the weekly Chl fields. We then subtract the resulting

from the original fields to produce $\delta$Chl fields. The $\delta$Chl fields are not sensitive to the selected $\sigma$. The choice of a rather small filter makes $\delta$Chl amplitudes smaller compared to if a larger filter is chosen, producing a conservative estimate of $\delta$Chl. In order to generate spatial maps of $\delta$Chl, we averaged all eddy associated anomalies of the respective eddy polarity into $5° \times 3°$ longitude/latitude boxes.

Prior to all analysis we log-transform Chl, due to Chl being lognormally distributed (Campbell, 1995). $\delta$Chl is frequently

given in percentage difference relative to the background Chl as

$$\delta\text{Chl} = \left[ \exp\left( \log(\text{Chl}_e)\text{-}\log(\text{Chl}_{bg}) \right) - 1 \right] \times 100 = \left( \frac{\text{Chl}_e}{\text{Chl}_{bg}} - 1 \right) \times 100$$

with subscripts e and bg denoting *eddy* and *background*, respectively. Where we show absolute $\delta$Chl on a logarithmic scale, we use the base 10 logarithm.

Regarding the spatial, i.e., geographical analysis, we use on the one hand the positions of the main ACC fronts (Polar Front,

PF, and Subantarctic Front, SAF) as determined by Sallée et al. (2008). On the other hand, we make use of a climatology of sea surface height (SSH) contours (Maximenko et al., 2009), which are representative for the long-term geostrophic flow in the area. The mean positions of the PF and SAF align approximately with the mean SSH contours of about -40 cm and -80 cm, respectively. We select the -20 cm SSH contour to separate waters of the southern subtropical gyres to the north of the ACC, referred to as subtropical waters from waters in the "ACC influence area", referred to as ACC waters. This choice is based

on both, a tendency for net eastward propagation of eddies south of this contour (Frenger et al., 2015) indicating advection by the ACC mean flow, and a seasonal sign switch of $\delta$Chl, which will be addressed later in the paper. Waters south of the PF/-80 cm SSH we refer to as Antarctic waters. Finally, we use mixed layer depths derived from Argo floats, available at http://www.locean-ipsl.upmc.fr.





### 2.3 Analysis of environmental Chl conditions

Using the data presented in the previous section, we calculate a monthly Chl climatology. Based on this climatology we derive *potential* $\delta$Chl ($\hat{\delta}$Chl) eddies may induce due to lateral advection (Figure 1a): in order to assess the $\hat{\delta}$Chl emerging from *stirring* in the Southern Ocean, we compute the climatological meridional Chl gradient at the spatial scale of individual eddies, here

taken as two eddy radii ($\hat{\delta}$Chl$_{\text{stir}}$). To assess $\hat{\delta}$Chl emerging from *trapping*, we estimate the Chl variation along individual eddies' pathways by computing the difference of the climatological Chl at the location of an eddy origin and the climatological Chl at the present location of the eddy at the present month ($\hat{\delta}$Chl$_{\text{trap}}$).

### 2.4 Analysis of the spatial shape of $\delta$Chl

We compute the composite spatial shape of Chl and $\delta$Chl associated with eddies the same way as done in Frenger et al. (2015)

for sea surface temperatures: we extract a squared subregion of side lenghts of 10 eddy radii for each individual eddy from the weekly maps of SLA and Chl, centered at the eddy center. We rotate Chl snapshots according to the ambient Chl gradient and average over all eddies to produce the eddy composite. Note that the estimate of the magnitudes of the dipole and the ambient Chl gradient (see below) tend to be slightly weaker without rotation. Nevertheless, as averages are taken over regimes of largely similar orientation of the ambient Chl gradient (see Discussion section 4), our conclusions do not depend on whether

we rotate snapshots or not.

Further, we decompose the eddy-induced spatial average $\delta$Chl pattern into a monopole (MP) and dipole (DP) pattern by first constructing the monopole by computing radial averages of $\delta$Chl around the eddy center, i.e., $\delta$Chl$(r)_{\text{MP}} = \overline{\delta\text{Chl}(r)}$, where $r$ is the distance from the eddy center. In the second step, we calculate $\delta$Chl$_{\text{DP}}$ as a residual, i.e., by differencing the monopole pattern from the total signal. Even though this residual approach captures in the dipole structure any non-monopole

pattern, experience has shown that the $\delta$Chl$_{\text{DP}}$ typically feature dipoles (Frenger et al., 2015). In the final step, we quantify the amplitudes of the monopoles and the dipoles, assess the contribution of the two components to the spatial variance of the total signal based on the sum of variances (var), i.e. var$(\delta Chl) = $var$(\delta\text{Chl}_{\text{MP}}) + $var$(\delta\text{Chl}_{\text{DP}})$, and compute the local Chl gradient at the scale of two eddy radii, as an estimate of the potential maximum contribution of *stirring* to $\delta$Chl.

### 2.5 Handling of measurement error and data gaps

As the error of the satellite retrieved Chl for each individual eddy can easily be as large as the anomaly, an individual eddy signal may be undetectable with in-situ measurements (Siegel et al., 2011). The significance of our results arises from the large number of analyzed eddies. A reduction of the sample size due to missing Chl data arising from frequent cloud cover in the SO may affect the significance: On average for 45% of the eddies' Chl data was entirely missing. Missing values due to cloud cover-only (leaving aside missing data due to the polar night) increase from 20% at 30°S to 60% at 65°S. Anticyclones exhibit

a higher percentage of data gaps than cyclones (47% versus 42%), which can be explained by the impact of their sea surface temperature anomalies on cloud cover (Park et al., 2006; Small et al., 2008; Frenger et al., 2013). We account for the issue





of missing data by spatio-temporally aggregating the data. The significance of the results at the 95% level is tested based on t-tests as the data are about normally distributed in the log-space.

## 3 Results

### 3.1 Imprint of eddies on Chl

#### 3.1.1 Mean imprint

Averaged across the entire Southern Ocean and all seasons, we detect a significant, although small mean imprint of eddies on Chl (Supplementary Figure S1) for both anticyclonic (warm-core, SLA lifting and thermocline deepening) and cyclonic (cold-core, SLA deepening and thermocline lifting) eddies. The overall mean $\delta$Chl associated with anticyclones is -4 %, while that for cyclones is of even smaller magnitude, i.e., +1 %. The distributions around these small means are very broad, however,

with many anticyclones and cyclones having both, positive or negative $\delta$Chl, depending on the region and time of the year. The long tails of the distributions, corroborated by visual inspection of the individual $\delta$Chl of eddies suggest anomalies that are substantially larger than the mean. Thus, it appears that by averaging the signals in time and space, a substantial amount of information is lost. As a consequence, it is more insightful to disentangle the signals and to examine the regional and seasonal variation of $\delta$Chl.

#### 3.1.2 Spatial variability of imprint

The maps of the annual mean imprint of cyclonic and anticyclonic eddies on Chl clearly supports this hypothesis of a strong regional cancellation effect (Figure 2). First, the regional mean signal associated with eddies is indeed much larger than suggested by the mean $\delta$Chl across the entire Southern Ocean. In fact, around a quarter of the analyzed grid cells have absolute $\delta$Chl larger than 10 %, and in a significant number of grid cells the mean absolute $\delta$Chl exceeds several tens of percent (Figure

2b,c). Second, the signals associated with eddies of either polarity vary in sign across the different regions with regions of strong enhancements bordering regions with strong reductions (see also Figure 1 in Gaube et al. 2014). In the broadest sense, the pattern is zonal in nature, reflecting the zonal nature of the climatological Chl distribution (Figure 2a).

For anticyclones, $\delta$Chl is clearly negative in subtropical waters and in the regions around the western boundary currents (Figure 2b). These prevailing negative $\delta$Chl are contrasted by mostly positive $\delta$Chl along the ACC. Cyclones have a largely

similar spatial pattern, but of opposite sign (Figure 2c): Prevailing positive $\delta$Chl in subtropical waters are opposed by a band of negative, yet weaker anomalies along the ACC. South of the ACC, the pattern of $\delta$Chl is spotty for anticyclones as well as cyclones, with anticyclones and cyclones featuring average positive and negative $\delta$Chl, respectively. In summary, SLA and $\delta$Chl are largely negatively correlated in subtropical waters north of the ACC, and positively correlated along the ACC.

A few exceptions break this mostly zonal picture. An exceptional area of negative $\delta$Chl for cyclones in subtropical waters

of the eastern Indian Ocean disrupts the zonal band of largely positive anomalies. Also, $\delta$Chl in coastal/shelf areas often are distinct from open-ocean $\delta$Chl. A clear signal emerges south and southwest off the Australian and South-American coasts,





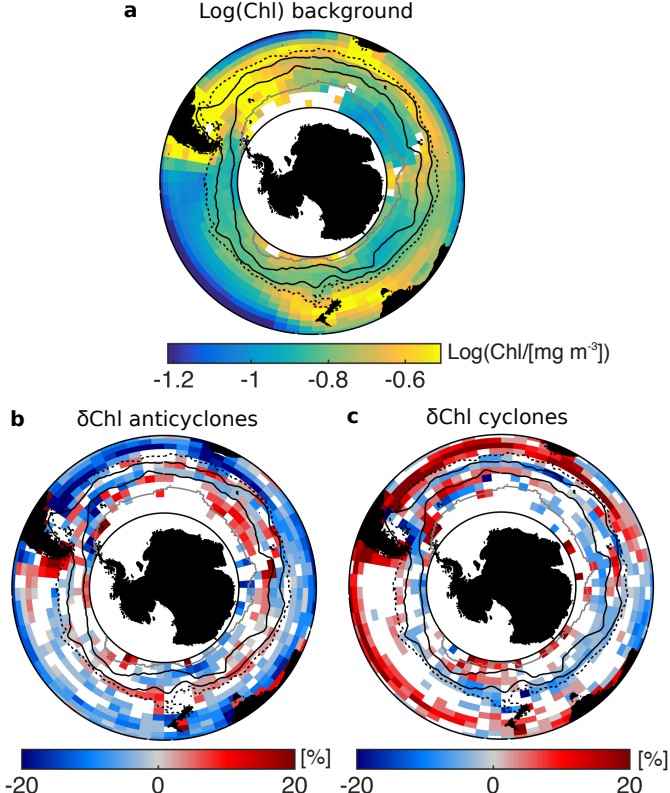

**Figure 2. Spatial distribution of chlorophyll anomalies ($\delta$Chl) associated with eddies; a** Logarithm (base 10) of climatological Chl for reference, and mean $\delta$Chl of **b** anticyclones and **c** cyclones; $\delta$Chl are the average of anomalies of eddies existing at least 3 weeks in $5° \times 3°$ longitude-latitude-grid boxes; white boxes indicate insufficient data or anomalies insignificantly different from zero (t-test, p=0.05); solid black lines mark the main branches of the ACC (Subantarctic and Polar fronts); the dashed black line denotes the -20 cm SSH contour and the solid gray line the northernmost extension of sea-ice cover.

west of New Zealand, and more subtly east of Kerguelen Islands and the Drake Passage (see also Sokolov and Rintoul 2007), where $\delta$Chl tends to be positive for both, anticyclones and cyclones.

### 3.1.3 Seasonality of imprint

The pronounced zonal bands of $\delta$Chl for anticyclones and cyclones persist over the year, but tend to migrate meridionally
5 (Figure 3a-d, middle/right columns), thereby following the pronounced seasonality of Chl (Figure 3a-d, left column; Thomalla et al. 2011; Sallée et al. 2015). The seasonality of $\delta$Chl is larger along the ACC and in Antarctic waters compared to subtropical waters. In the subtropical gyres, $\delta$Chl of anticyclones and cyclones are negative and positive, respectively, i.e., SLA and $\delta$Chl are negatively correlated all year round. Here, $\delta$Chl shows a weak peak in austral summer when climatological Chl is smallest (Figure 3c). In the ACC regions and in Antarctic waters, a striking feature is the seasonal change in the sign of $\delta$Chl (Figure



3b-d).

This becomes even more evident when inspecting the zonally averaged Chl and $\delta$Chl as a function of season and SSH, i.e., plotted in the form of a Hovmoeller diagram (Figure 4). Along the ACC, anticyclones exhibit negative $\delta$Chl in winter to spring concurrent with deep mixed layers, followed by positive $\delta$Chl in summer to autumn (Figure 4b). Cyclonic $\delta$Chl patterns are

opposite, featuring negative $\delta$Chl in spring to autumn, with close to zero to positive $\delta$Chl in winter to spring (Figure 4c). This implies that SLA and $\delta$Chl are positively correlated summer to autumn, followed by a negative correlation in winter to spring. This sign switch of the correlations shows a seasonal lag towards Antarctic waters, with positive correlations prevailing autumn to winter, and negative correlations prevailing spring to summer, resulting in the aforementioned apparent southward migration of the sign switch of the seasonality of $\delta$Chl over the course of the year.

## 3.2   Causes for the imprint

### 3.2.1   Advection

To assess if advective mechanisms are sufficient to explain the observed $\delta$Chl associated with eddies in the Southern Ocean, we contrast the observed changes against the advective *potentials*, i.e., the potential $\hat{\delta}$Chl$_{\text{stir}}$ associated with *stirring* and the

potential $\hat{\delta}$Chl$_{\text{trap}}$, associated with *trapping* (Figures 4d-g, Method section 2.3). If the patterns and magnitudes match, one may conclude that Southern Ocean eddies cause $\delta$Chl merely through lateral advection of Chl. We begin with a discussion of the potential of *stirring*.

In the northern domain, i.e., in subtropical waters (here SSH larger -20 cm) the sign of $\hat{\delta}$Chl$_{\text{stir}}$ tends to agree with $\delta$Chl throughout the year for both anticyclones and cyclones (Figures 4b-e). So does the seasonal variation of the magnitude of

$\hat{\delta}$Chl$_{\text{stir}}$, with the largest magnitudes found in summer to autumn. Also the regional variations match, such as a weaker $\hat{\delta}$Chl$_{\text{stir}}$ and $\delta$Chl in the Pacific sector compared to the Atlantic and Indian Ocean sectors (Figure 3, middle/right columns and Supplementary Figure S2, left column).

Also, along the ACC and its northern flank in summer to autumn, $\hat{\delta}$Chl$_{\text{stir}}$ and $\delta$Chl agree in sign, and are of the same order of magnitude. Finally, along the southern ACC and in Antarctic waters, $\hat{\delta}$Chl$_{\text{stir}}$ mirrors the seasonal sign switch of $\delta$Chl, and

the apparent seasonal southward migration of the zonal bands of $\delta$Chl (Figures 3 and 4b,c). Thus, it appears that *stirring* can already explain a good fraction of the observed $\delta$Chl (i) in subtropical waters outside of those characterized by winter deep mixed layers, (ii) along the ACC and its northern flank in summer to autumn, and (iii) south of the ACC.

That *stirring* has the potential to produce the observed $\delta$Chl in large parts due to a sufficiently large average ambient gradient of Chl, is corroborated by the actual observed local shape of $\delta$Chl (Method section 2.4). For instance: averaged over eddies

in subtropical waters in the northern domain in winter to spring, the average absolute gradient at scales of two eddy radii is 7 % for both anticyclones and cyclones, and the maximum $\delta$Chl -10 % and 9 %, respectively (Figure 5a, see numbers at the bottoms of left two panels). Similarly, along the ACC and its northern flank (Figure 6a), the average Chl gradient at the scale of eddies in summer to autumn, and in Antarctic waters in spring (Figure 5b), are of similar magnitudes as $\delta$Chl, supporting





**Figure 3. Seasonality of chlorophyll anomalies (δChl) associated with eddies;** Austral **a** winter, **b** spring, **c** summer and **d** autumn for anticyclones (middle) and cyclones (right); logarithm (base 10) of climatological Chl for reference (left). Otherwise as Figure 2.





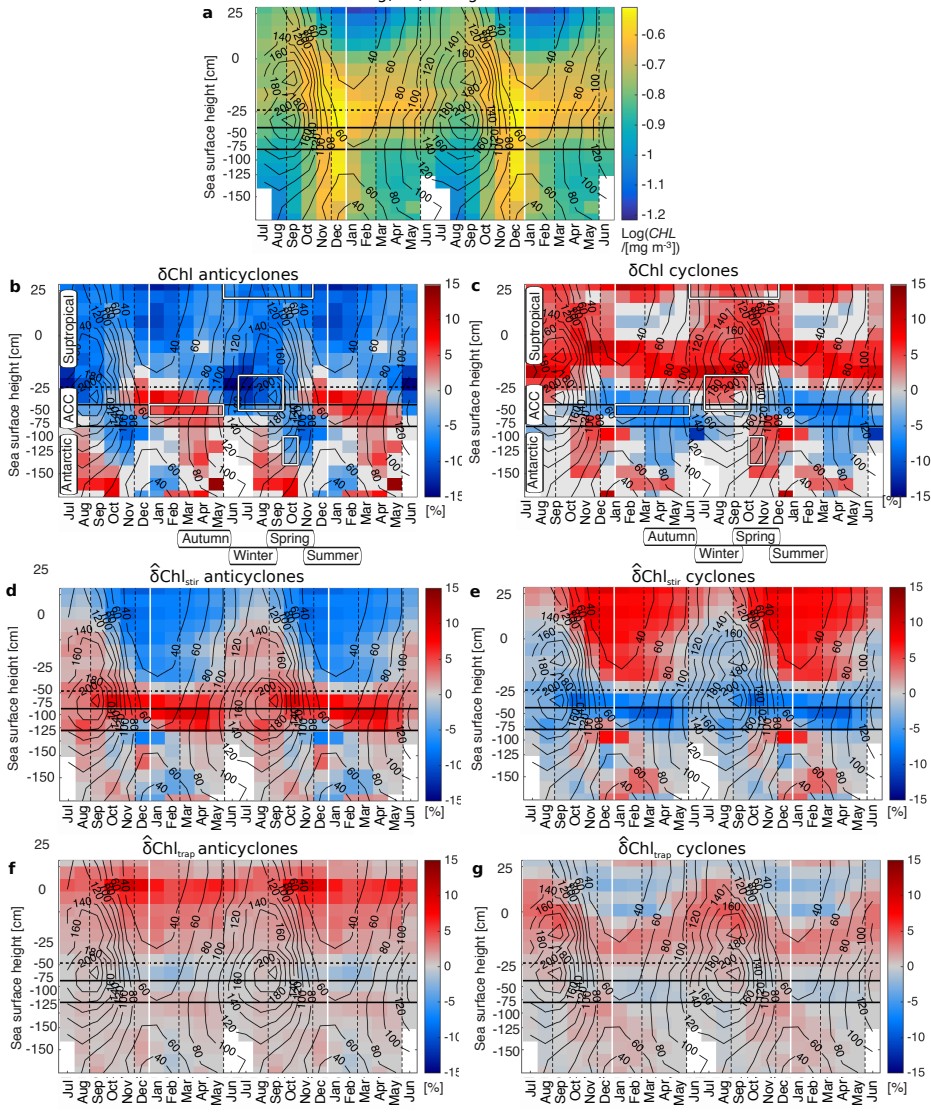

**Figure 4. Seasonality of chlorophyll anomalies ($\delta$Chl) associated with eddies, and potential of eddies to cause $\delta$Chl through lateral advection ($\hat{\delta}Chl$); a** Base 10 logarithm of climatological Chl for reference, and $\delta$Chl related to **b** anticyclones and **c** cyclones; $\delta$Chl are the mean of all eddies existing at least 3 weeks binned in monthly sea surface height (SSH) bins so that boxes roughly are of equal area; $\delta$Chl that are not significant (t-test, p=0.05) are colored in light gray, missing data in white; solid black lines mark the ACC (approximate positions of the Subantarctic and Polar fronts); the horizontal dashed black line denotes the -20 cm SSH contour, the vertical dashed lines seasons; solid black contours show averaged mixed layer depths; note that the seasonal cycle is shown repeatedly to highlight cyclic patterns; advective potentials (Method section 2.3) due to **d,e** *stirring* and subsequent lateral entrainment/permeable *trapping*, by anticyclones and cyclones, respectively; **f,g** advective potential of *trapping*, for anticyclones and cyclones, respectively; i.e. the case for idealized impermeable eddies with associated efficient material trapping in the core and constant Chl levels; see text for the method to estimate the potentials.





that *stirring* alone may largely explain observed $\delta$Chl (Figure 6a; anticyclones: gradient of 9 % and maximum $\delta$Chl of 5 %; cyclones: gradient of 9 % and maximum $\delta$Chl of -11 %; and Figure 5b; anticyclones: gradient of 5 % and maximum $\delta$Chl of -6 %; cyclones: gradient of 5 % and maximum $\delta$Chl of 5 %).

The advective potential for the other lateral advective mechanism, i.e., trapping, $\hat{\delta}$Chl$_{\text{trap}}$, partly opposes and partly enhances

$\hat{\delta}$Chl$_{\text{stir}}$ (Figures 4d-g). For instance, for cyclones along the ACC in summer to autumn, *trapping* possibly contributes to a $\delta$Chl (11 %) signal that is slightly larger than the Chl gradient at two eddy radii (9 %), and the contribution of the variance of the monopole is increased compared to anticyclones (Figure 6a, 96 % versus 87 %). Yet, overall the trapping potential $\hat{\delta}$Chl$_{\text{trap}}$ is weak compared to $\delta$Chl (Figure 4b,c,f,g), and outweighed by $\hat{\delta}$Chl$_{\text{stir}}$.

### 3.2.2   Biogeochemical rates

Even though advective processes and particularly *stirring* appear to be the dominant driver for the eddy-associated chlorophyll anomalies, there are nevertheless a few places where the magnitudes of the potentials for advective effects are too weak compared to the observed $\delta$Chl or of opposite sign. These are the places where variations in the biogeochemical rates may be the dominant driver.

The most prominent instance is found along the northern ACC associated with the seasonal sign switch of $\delta$Chl (Figures 4b-g,

blue boxes in Figure 7a). Here, anticyclones switch to negative $\delta$Chl in the presence of deep mixed layers whereas both, $\hat{\delta}$Chl$_{\text{stir}}$ and $\hat{\delta}$Chl$_{\text{trap}}$ suggest positive $\delta$Chl. The shape of the local imprint of anticyclones in the respective region and season (Figure 6b) indicates that indeed, the lateral Chl gradient is small at the scale of eddies (5 %) compared to the maximum absolute amplitude of $\delta$Chl (17 %). Further, a decomposition of the local shape of $\delta$Chl suggests that *stirring* causes an anomaly of the opposite sign than $\delta$Chl, consistent with Figure 4d. Given that *trapping* similarly would cause a weak anomaly of the opposite

sign (Figure 4f), we hypothesize that eddy-induced changes in the biogeochemical rates are responsible for the negative $\delta$Chl in winter and spring in the northern ACC.

Similarly, the sign switch of $\delta$Chl of cyclones in the same region cannot be explained based on $\hat{\delta}$Chl$_{\text{stir}}$ (Figure 4e). The local shape of Chl corroborates that also for cyclones *stirring* in the average ambient Chl gradient induces an anomaly of the opposite sign (Figure 6b). In contrast to $\hat{\delta}$Chl$_{\text{stir}}$, $\hat{\delta}$Chl$_{\text{trap}}$ for cyclones is of the same sign as $\delta$Chl (Figure 4g), indicating

a potential contribution of *trapping* to positive $\delta$Chl under deep mixed layers. Yet, as noted in the previous paragraph, the magnitude of $\hat{\delta}$Chl$_{\text{trap}}$ is small, hence the contribution by *trapping* is limited. Further, *trapping* is not of the same sign as $\delta$Chl for cyclones everywhere in the region either (see blue boxes Figure 8a, right column). Hence, an effect of eddies on biogeochemical rates is required also for cyclones in winter deep mixed layers.

Effects of eddies on biogeochemical rates may play a role in other regions or seasons, too. For instance the magnitudes of

$\hat{\delta}$Chl$_{\text{stir}}$ and $\hat{\delta}$Chl$_{\text{trap}}$ appear too weak to explain $\delta$Chl in subtropical waters in winter and spring (Figures 4d-g/5a). Further, the potential $\hat{\delta}$Chl$_{\text{trap}}$ is too weak to explain the strong monopole component of $\delta$Chl (Figures 4f,g/5/6): The local shape of Chl shows closed Chl contours associated with eddies despite the weak $\hat{\delta}$Chl$_{\text{trap}}$ (Figures 5/6, left columns), suggesting an effect on biogeochemical rates that may enhance the $\delta$Chl monopole.



**Figure 5. Attribution of *stirring/trapping* components**; average eddy chlorophyll (Chl) in regions **a** R1 (SSH larger 10 cm, June to November) and **b** R4 (SSH -140 to -100 cm, March), indicated as white boxes in Figures 4b,c and 7; for anticyclones (top rows) and cyclones (bottom rows); (left column) shows the logarithm (base 10) of Chl and (second left) $\delta$Chl (stippling marks insignificant anomalies), decomposed into (second right) a monopole MP and (right) the residual (approximately dipole DP) contribution (see text for details and cartoon in Figure 1); sea level anomaly contours are shown in black (0.05 spacing, normalized before averaging); the inner and outer white circles indicate the eddy core and area used for the computation of the contribution to the variance of $\delta$Chl of the monopole and the dipole, respectively; text in panels denotes (left) the meridional Chl gradient at two eddy radii, (second left) the maximum or minimum of the anomaly, (second right and right) the contribution to the variance of the anomaly pattern of the monopole and dipole, respectively; before averaging, the individual eddy snapshots are scaled according to the eddy's radius (R) and rotated according to the ambient instantaneous Chl gradient.





**Figure 6. Attribution of *stirring/trapping* components**; same as Figure 5 but for **a** autumn ACC waters R2 (SSH -60 to -40 cm, January to May) and **b** R3 northern ACC winter deep mixed layer waters (SSH -50 to -15 cm, July to September); average regions are indicated with boxes in Figure 7.







**Figure 7. Regions of potential of eddies ($\hat{\delta}$Chl) to cause chlorophyll anomalies ($\delta$Chl)**; due to *stirring*, *trapping*, a combination of the two or non of the two; for **a** anticyclones and **b** cyclones; legend: sign of eddy induced $\delta$Chl agrees with sign of potential *Trapping* effect, *Stirring* effect, with the sign of both (*Trapping and stirring*), or with the sign of none of the two (*Neither trapping nor stirring*); see text for details; white boxes indicate regions R1 to R4 used for composite Figures 5/6.







**Figure 8. Seasonality of potential of eddies ($\hat{\delta}$Chl);** Austral **a** winter, **b** spring, **c** summer and **d** autumn for anticyclones (left) and cyclones (right). Otherwise as Figure 2.





## 4   Discussion and synthesis

The zonal pattern of the eddy induced Chl anomalies, $\delta$Chl, identified here for the Southern Ocean is similar to that seen in global ocean-based analyses (Gaube et al., 2014). Also the magnitude of $\delta$Chl is similar north of the ACC to what Gaube et al. (2014) reported on a global basis. Yet, along the ACC we find more widespread and more intense $\delta$Chl than the global study,

especially in summer and autumn. *Regional* variations in the sign of $\delta$Chl associated with either cyclonic and anticyclonic eddies have been reported previously (Gaube et al., 2014). Such regional variations are considerable also in the Southern Ocean. In contrast, *seasonal* variations have been reported to be relatively weak globally (Gaube et al., 2014), except for the eastern Indian Ocean and the South China Sea (Gaube et al., 2013; Guo et al., 2017). And seasonal changes in the sign of $\delta$Chl in a particular region have to our knowledge not been reported before. Hence, the strong seasonality with a seasonal change in

the sign of $\delta$Chl along the ACC and south of the ACC appears to be rather specific to the Southern Ocean.

The spatial and seasonal variability of $\delta$Chl may not be that surprising in hindsight, given that the same mechanism, e.g., advection can lead to either positive or negative signs for the same polarity depending on the sign of the lateral gradient. In addition, several mechanisms may be involved simultaneously, so that small differences in their relative importance can lead to substantial differences in the net sign of the response (Siegel et al., 2011; Gaube et al., 2014; McGillicuddy, 2016). In

the end, we demonstrated that most of the eddy induced signatures of $\delta$Chl in the Southern Ocean are likely due to *stirring*, a mechanism that has been shown to control $\delta$Chl in the low to mid-latitude ocean as well (Chelton et al., 2011a). But we showed also that there are several other regions/seasons where other processes, namely trapping and changes in biogeochemical rates appear dominant. To illustrate this, we took the Hovmoeller diagram of Figure 4 and identified the dominant mechanism based on the results of the analyses of the two advective potentials (Figure 7).

This synthesis figure (Figure 7) reveals that the dominance of *stirring* as the sole mechanism is limited to the subtropical waters outside of deep mixed layers, and for anticyclones along the northern ACC in summer to autumn (Figure 7, yellow). Our results suggest that *trapping* contributes to $\delta$Chl for anticyclones along the southern ACC in summer to autumn and in Antarctic waters in autumn and spring, and to $\delta$Chl of cyclones in most regions and seasons, except for subtropical waters in winter to spring (see also Figure 8a, south and southwest of Australia). Yet, the magnitude of the potential of *trapping* is generally

weak, with the exception, perhaps, of a few specific regions, such as by eddies originating from eastern boundary currents (Supplementary Figure S3). They tend to move westward across transitional regions of coastal to off-shore waters, i.e., down intense Chl gradients (Supplementary Figure S2, right column), with a resulting positive $\delta$Chl, e.g., in the southeast Pacific or southeast of Kerguelen Islands and Drake Passage. In these regions, $\delta$Chl is positive year round for both anticyclones and cyclones (Figure 3). A possible explanation next to advection of Chl is the offshore advection of iron trapped in the nearshore

region by eddies that fuels extra growth in the offshore waters, as suggested e.g., for Haida eddies in the North Pacific (Xiu et al., 2011). A substantial effect of *trapping* to cause $\delta$Chl in boundary currents corroborates previous results (Gaube et al., 2014).

The nevertheless *overall* weaker role of *trapping* relative to *stirring* is consistent with (i) a propagation distance of eddies over their lifetime that is on average smaller than two eddy radii, meaning that the scale of impact due to eddy propagation tends





to be smaller than the one due to eddy rotation, and (ii) an inherently westward propagation of eddies, meaning a propagation largely along Chl isolines, as zonal Chl gradients typically are much smaller than meridional Chl gradients.

Moreover, a weak *trapping* signal also is anticipated as trapped waters from the eddies' origins will be diluted along the eddies' pathways. Eddies tend to not trap perfectly but continuously leak and entrain waters ambient to their cores (Beron-Vera
et al., 2013; Wang et al., 2015; Haller, 2015). The importance of such lateral entrainment or permeable *trapping* is supported by the pronounced monopole contributions to the shapes of the local $\delta$Chl imprint (Figures 5,6). The dipole contributions are relatively weak despite the comparatively steep ambient Chl gradients favoring *stirring*. In summary we hypothesize that the relative weakness of the dipoles stems from lateral entrainment of Chl into the eddies' cores, resulting in a pronounced asymmetry of the dipoles and a large monopole part in the $\delta$Chl decomposition (see illustration in Figure 1a).

The clearest case for a substantial contribution of changes in biogeochemical rates on $\delta$Chl was found for the northern ACC region during winter and spring, when the mixed layers are deep (Figure 7, blue), and correlations of Chl and SLA are negative. The associated negative $\delta$Chl of anticyclones is consistent with the mechanism of an amplification of large-scale prevailing light limitation in deep mixed layers (Boyd, 2002; Moore and Abbott, 2002; Venables and Meredith, 2009; Fauchereau et al., 2011): Anticyclones tend to deepen isopycnals, causing deeper mixed layers of several tens of meters and weaker mixed layer
stratification, especially in winter (Song et al., 2015; Hausmann et al., 2017; Dufois et al., 2016). Hence, phytoplankton within the mixed layer will be exposed to reduced mean radiation in anticyclones as compared to ambient waters, and vice versa for cyclones.

Our result of a pronounced monopole-shape of $\delta$Chl despite the weak trapping potential suggests that also in summer to autumn, positive correlations of SLA and $\delta$Chl are at least partly caused by effects of eddies on biogeochemical rates. Here,
prevailing iron limitation could be modulated by eddies, with an abatement of the iron limitation and associated positive $\delta$Chl caused by weakly stratified anticyclones in high wind conditions and associated intensified vertical mixing, and vice versa for cyclones (Boyd et al., 2012; Dufois et al., 2016; Song et al., in revision). Moreover, alleviation of grazing pressure due to reduced predator-prey encounter rates in deepened mixed layers in anticyclones could favor positive $\delta$Chl, again vice versa for cyclones. Thus, we argue that along the northern ACC, the seasonal sign switch of $\delta$Chl could be explained by varying
degrees of light and iron limitation and grazing pressure over the course of the year (Boyd, 2002; Venables and Meredith, 2009; Carranza and Gille, 2015; Le Quéré et al., 2016).

Finally, along the southern ACC and in Antarctic waters in autumn to spring, the potentials of *stirring* and *trapping* often-times are of the same sign. However, $\delta$Chl associated with eddies is insignificant (dark gray regions, Figure 7). Presumably, these situations when $\delta$Chl are insignificant arise because eddy effects on biogeochemical rates oppose advective effects.

We note that our analysis is constrained to the surface ocean, hence three aspects need to be kept in mind: (i) one potential issue are non-homogeneous vertical Chl profiles, e.g., the presence of unrecognized subsurface Chl maxima. As in previous studies (Sallée et al., 2015), we assume that in our focus region, at the core latitudes of the Southern Ocean across the ACC, subsurface Chl maxima are not prominent as wind speeds are high and mixed layers deep, promoting well-mixed Chl levels over the upper ocean; further, surface and mixed layer depth integrated analyses provide similar results in terms of SLA-Chl
correlations (based on model simulations, Hajoon Song, pers. communication), supporting the feasibility of an analysis of





surface Chl. (ii) Modification of mixed layer depths by eddies may result in a surface Chl concentration modification due to a dilution effect. Especially in winter to spring when the mixed layers are deep we cannot exclude that this effect contributes to negative and positive $\delta$Chl for anticyclones (deepening thermocline) and cyclones (lifting thermocline), respectively. Yet as noted in (i), surface and mixed layer depth integrated analyses provide similar results in a model simulation. Further, given the

considerable spatio-temporal variation of $\delta$Chl across the Southern Ocean dissimilar to mixed layer depth variations, this is likely not the major factor responsible for $\delta$Chl. Finally , (iii) potential effects of eddies on phytoplankton growth presumably occur in the lower euphotic zone and may be expressed more weakly at the surface where they are detected by satellite sensors (McGillicuddy et al., 2007; Siegel et al., 2011). We therefore note that effects of eddies on biogeochemical rates may be underestimated in this surface based study.

## 5   Summary and Conclusions

The prevalent and strong correlations between anomalies in surface Chlorophyll and mesoscale variability have triggered substantial research, but many unresolved issues remain, particularly associated with the question of their causes (Lévy, 2008; Gaube et al., 2014; McGillicuddy, 2016). With this study, we aim to provide an observational reference for the seasonal climatological $\delta$Chl associated with mesoscale eddies across the Southern Ocean, a region where the detailed regional and

seasonal relationship of eddies and Chl previously had not been considered. We have obtained the estimate by combining satellite estimates of Chl with ocean eddies identified based on satellite estimates of SLA. A large number of collocations of eddies and Chl allowed us to retrieve statistically robust results despite frequent data gaps and high spatio-temporal variability of Chl.

We found $\delta$Chl associated with eddies of >10% over wide areas in the Southern Ocean. The large-scale patterns are positive

and negative anomalies for cyclones in subtropical waters and along the ACC, respectively; anticyclones show a similar pattern, but of opposite sign. A pronounced seasonality of the imprint is apparent especially along the ACC and in Antarctic waters, featuring a sign switch of the anomaly over the course of the year.

While multiple mechanisms may be at play at the same time to cause $\delta$Chl (Gaube et al., 2014; McGillicuddy, 2016), our analyses based on climatological Chl gradients, eddy rotation and propagation pathways, and the local shape of $\delta$Chl of eddies

lead us to conclude that lateral advection due to *stirring* by eddies and associated lateral entrainment and permeable *trapping* have the potential to explain a large fraction of Southern Ocean eddy-induced $\delta$Chl.

A prominent region and season where eddy-induced advection is insufficient to explain $\delta$Chl are the northern ACC characterized by deep mixed layers in winter to spring and the seasonal sign switch of $\delta$Chl in the same region: Here, winter to spring negative and positive $\delta$Chl of anticyclones and cyclones, respectively, are consistent with an enhancement and reduction of

deep mixed layer light limitation. The opposite signs of $\delta$Chl in summer to autumn are consistent with an abatement of grazing pressure caused by anticyclones via deepened mixed layers, or iron limitation via a relatively weak stratification facilitating vertical mixing, and vice versa for cyclones. In other regions and seasons our analysis does not exclude a modulation of $\delta$Chl



by effects of eddies on biogeochemical rates, even though our results suggest that lateral advection has the potential to be the dominant mechanism.

Future work may include to further investigate the extent of lateral entrainment and permeable *trapping* of eddies versus non-local *trapping*, and to test where and when Southern Ocean eddies substantially affect biogeochemical rates, such as

through modulation of alternating roles of iron and light limitation and grazing pressure along the ACC. The growing number of sub-surface biogeochemical measurements across eddies may be of help, collected by biogeochemical floats, gliders and seals. In addition, targeted experiments with numerical ocean-biogeochemical models with the option to alternately switch on and off Chl sources-sinks would be useful to shed light on the questions, of what the role of eddy-effects is on Chl sources-sinks relative to advection, for higher trophic levels (Xiu et al., 2011; Nel et al., 2001; Godø et al., 2012), or for the intensity of export

(Waite et al., 2016). Furthermore, numerical models would allow us to assess if these effects of eddies on Chl substantially affect Southern Ocean biogeochemistry, such as of mode and intermediate waters which form from winter deep mixed layers, supply low latitude ocean with nutrients and sequester anthropogenic carbon (Sarmiento et al., 2004; Sallée et al., 2012).

*Data availability.* The identified eddies we use in this study including their Chl characteristics are publicly available (http://dx.doi.org/10.3929/ethz-b-000238826). Other presented data are available from the corresponding author upon request.

*Author contributions.* I.F., N.G., and M.M. conceived the project, I.F. carried out the analyses, all authors contributed to the writing of the manuscript.

*Competing interests.* The authors report no competing financial interests.

*Acknowledgements.* The altimeter products used for this study were produced by Ssalto/Duacs and distributed by Aviso, with support from Cnes (http://www.aviso.oceanobs.com/duacs/). The $\delta$Chl used were processed and distributed by ACRI-ST GlobColour service, supported

by EU FP7 MyOcean & ESA GlobColour Projects, using ESA ENVISAT MERIS data, NASA MODIS and SeaWiFS data. QuikScat data are produced by Remote Sensing Systems and sponsored by the NASA Ocean Vector Winds Science Team. Data are available at http://www.remss.com.





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
