# Peer review of "Imprint of Southern Ocean mesoscale eddies on chlorophyll"

_Biogeosciences, 2018_

## Referee Comment (RC1) · F. d'Ovidio (Referee) · 16 Mar 2018

The paper provides an analysis of the role of mesoscale eddies in the Southern Ocean on primary production, in terms of chlorophyll anomalies detected by remote sensing associated to them. Methodologically, the paper follows very closely some previous works, in particular Gaube et al. 2014, which were more focused on the global ocean. In respect to previous works, this manuscript has fine tuned the methodology, and discussed the results in the specific context of the Southern Ocean. The main original result in this manuscript is the finding of a strong seasonal signal in the mesoscale imprint on chlorophyll anomalies. This result and other more incremental findings are not surprising, but are very well discussed in terms of the previous literature and in terms of the biogeochemical activity of the Southern ocean (with possibly one direction of improvement which is described below). As a consequence, I find this manuscript as a useful contribution to the understanding of the role of mesoscale eddies on primary

production in the Southern Ocean, even in its current form. There is however one issue that may improve further the manuscript.

One of the main concept treated by the paper, is stirring, and in particular the imprint of stirring induced by the mesoscale eddies on the mesoscale anomalies of the chlorophyll field. The manuscript explains that the stirring created by a mesoscale eddy can create a local deformation of a pre-existing chlorophyll gradient and I agree with this statement. However, this is not all about stirring. In fact, if I think to the imprint of stirring and chlorophyll in the Southern ocean, the main effect that comes to my mind is not the generation of local chlorophyll anomalies, but the huge plumes of chlorophyll induced when stirring by mesoscale eddies modulates iron delivery in a non-local way, preconditioning the blooms of this region. An analysis of this effect is not in the scope of this paper, and it has been done elsewhere (for instance, d'Ovidio et al. Biogeosciences 12, 2015; Ardyna et al. GRL 44, 2017). Nevertheless, I feel that the submitted manuscript should stress more that what the authors intend here for eddy stirring, is only the local effect of stirring, while other non-local effects of stirring by mesoscale activity also exist, and actually they are a prominent driver of the bloom extension and intensity in the Southern Ocean. In fact, it would be interested to know whether there is a signature of this non-local stirring effect in the analysis presented, for instance by finding stronger anomalies downstream of likely iron sources like the continental shelves present in the region. Or as a possible alternative explanation of the asymmetries in the chlorophyll anomalies.

I am certainly biased in this comment by my own work on the subject, therefore the authors are free to find some other papers instead of the two indicated above to add to their discussion. But in any case, I feel that the discussion on stirring merits to be extended.

Minor comments

timescale of chlorophyll: chlorophyll is just a pigment. Referring to the timescale of a

bloom, or of phytoplankton demography, should be more appropriate.

---

## Referee Comment (RC2) · V. Strass (Referee) · 29 Mar 2018

General Appraisal

The paper presents the results of a truly impressive data analysis of the effects of mesoscale eddies on sea surface chlorophyll in the Southern Ocean, comprising an extensive and widely new look into the regional and seasonal variation of these effects. In respect of how those interesting results are set into scientific context, however, the paper has severe weaknesses. I think these weaknesses can be overcome by rewriting major parts of the manuscript, sections 1, 4 and 5 in particular.

Specific Comments

The mesoscale ocean dynamics govern the range from a few kilometres to a few hundreds of kilometres horizontally. The data used in the study by Frenger et al., col-

lected by satellite remote sensing, provide a horizontal resolution of 1/3 of a degree for eddies (Aviso SLA) and 0.25° for the concentration of sea surface chlorophyll (ESA GlobColour Project product), i.e. approx. 37 km and 28 km in latitude, respectively. In consequence, only the larger fraction of mesoscale eddies is investigated. This needs, but is not yet, be made clear in the paper.

Eddies, or the mesoscale dynamics in general, affect phytoplankton hence the chlorophyll concentration in various ways, particularly by time-variable horizontal and vertical advection and associated transports of nutrients, and by vertical current shears that control stratification and subduction hence the light environment which phytoplankton cells experience. (In the Southern Ocean, where most macro-nutrients are abundant, it is likely the mesoscale upwelling of the primary production-limiting micro-nutrient iron that enhances biological production in the ACC with its meandering fronts; Hense, et al., Regional ecosystem dynamics in the ACC: Simulations with a three-dimensional ocean-plankton model , J. Mar. Systems, 2003.) Vertical velocities, and therefore possible upwelling of nutrients, but are known be most intense at the smallest part ($\leq$ 10 km) of the mesoscale range (Martin et al., Patchy productivity in the open ocean, Global Biogeochem. Cycles 2002; Lévy, Mesoscale variability of phytoplankton and of new production: Impact of the large‐scale nutrient distribution, J. Geophys. Res., 2003; Klein & Lapeyre, The oceanic vertical pump induced by mesoscale and sub-mesoscale turbulence, Annu. Rev. Mar. Sci. 2009). The relevance of those small scales has been noted initially by Woods (Mesoscale upwelling and primary production, in Toward a theory on biological–physical interactions in the world ocean, ed. B. J. Rothschild, Dordrecht Kluwer,1988), who raised the hypothesis that key to understanding the plankton patchiness which was revealed with the advent of satellite chlorophyll images, lies in the dynamics of mesoscale jets, where dynamical constraints limit upwelling to horizontal dimensions of about ten kilometres. This hypothesis received first observational support in 1992 (Strass, Chlorophyll patchiness caused by mesoscale upwelling at fronts, Deep-Sea Res. I). These latter two publications, by the way, would close the glaring time gap of the literature review given in the Introduction (p.1, lines 20

– 22.) between the cited advent of satellite chlorophyll images (Gower et al. 1980) and Doney (2003).

The presumably most important horizontal scale for stimulating phytoplankton growth, as explained above, unfortunately is not resolved by the present study. Moreover, most of the above-mentioned studies have demonstrated that up- and downwelling predominately are driven by changes in time of the mesoscale flow field (related to the development of frontal meanders due to baroclinic instability, frontogenesis by eddy-eddy interaction etc.). For the ACC, Strass et al. (Mesoscale frontal dynamics: Shaping the environment of primary production in the Antarctic Circumpolar Current, Deep-Sea Research II, 2002) have shown with an in situ study that the acceleration/deceleration of a frontal jet by interaction with an eddy creates a pattern of up- and downwelling cells and of chlorophyll patches on a much smaller horizontal scale than that of the involved eddy. Frenger et al. in their present study, however, analysed only eddies that were tracked over at least three weeks, hence eddies which were not subject to much change in time. Both by the selection of eddies of larger size and of low temporal change, Frenger and co-authors likely introduce a bias towards eddies of rather limited impact on biological production and biogeochemical rates. Their conclusion that eddy-driven stirring and trapping dominate over biogeochemical effects therefore seems not robust but rather a result of the horizontal/time scale bias. This requires an honest and thorough discussion.

Throughout their ms Frenger and co-authors associate cyclonic eddies with thermo-cline lifting and anticyclonic eddies with thermocline deepening. Undisputable is that cyclones display a lifted thermocline and anticyclones a deepened thermocline. However, whether or not the thermocline moves up or down after eddies have been fully formed is in contestation. It may well be the reverse of the indicated way, i.e. that during eddy slow-down due to processes such as eddy-induced Ekman pumping, the thermocline in cyclones moves downward and in anticyclones upward (e.g. Gaube et al., 2014). I therefore recommend the authors use to a more careful wording, i.e.

lifted/deepened instead of lifting/deepening.

On p. 20, lines 30-31 the authors bring forward the argument that anticyclones cause an abatement of grazing pressure, without providing a reference. In general I would doubt that a reference for this argument exists, which could be considered representing the widely accepted and unquestioned state of knowledge regarding mesoscale variability of grazing. Therefore, I consider this argument pure speculation, and suggest remove it from the ms.

Technical Issues

Fig. 4 should be enlarged to full-page size to enhance its readability in the pdf-version of the paper, and the caption therefore be shifted to next page, if possible.

Caption Fig 6 associates autumn with the months January to May, what is certainly not correct. If the given months are valid, then the season should be termed high summer – autumn or so.

---

## Author Comment (AC1) · 18 May 2018

Reviewer Comment 1, F. d'Ovidio, and *author response*:

**Major comments**

The paper provides an analysis of the role of mesoscale eddies in the Southern Ocean on primary production, in terms of chlorophyll anomalies detected by remote sensing associated to them. Methodologically, the paper follows very closely some previous works, in particular Gaube et al. 2014, which were more focused on the global ocean. In respect to previous works, this manuscript has fine tuned the methodology, and discussed the results in the specific context of the Southern Ocean. The main original result in this manuscript is the finding of a strong seasonal signal in the mesoscale imprint on chlorophyll anomalies. This result and other more incremental findings are not surprising, but are very well discussed in terms of the previous literature and in terms of the biogeochemical activity of the Southern ocean (with possibly one direction of improvement which is described below). As a consequence, I find this manuscript as a useful contribution to the understanding of the role of mesoscale eddies on primary production in the Southern Ocean, even in its current form.

*Thank you for the supportive review.*

**Comment a**

1. There is however one issue that may improve further the manuscript. One of the main concept treated by the paper, is stirring, and in particular the imprint of stirring induced by the mesoscale eddies on the mesoscale anomalies of the chlorophyll field. The manuscript explains that the stirring created by a mesoscale eddy can create a local deformation of a pre-existing chlorophyll gradient and I agree with this statement. However, this is not all about stirring. In fact, if I think to the imprint of stirring and chlorophyll in the Southern ocean, the main effect that comes to my mind is not the generation of local chlorophyll anomalies, but the huge plumes of chlorophyll induced when stirring by mesoscale eddies modulates iron delivery in a non-local way, preconditioning the blooms of this region. An analysis of this effect is not in the scope of this paper, and it has been done elsewhere (for instance, d'Ovidio et al. Biogeosciences 12, 2015; Ardyna et al. GRL 44, 2017). Nevertheless, I feel that the submitted manuscript should stress more that what the authors intend here for eddy stirring, is only the local effect of stirring, while other non-local effects of stirring by mesoscale activity also exist, and actually they are a prominent driver of the bloom extension and intensity in the Southern Ocean. In fact, it would be interested to know whether there is a signature of this non-local stirring effect in the analysis presented, for instance by finding stronger anomalies downstream of likely iron sources like the continental shelves present in the region. Or as a possible alternative explanation of the asymmetries in the chlorophyll anomalies. I am certainly biased in this comment by my

own work on the subject, therefore the authors are free to find some other papers instead of the two indicated above to add to their discussion. But in any case, I feel that the discussion on stirring merits to be extended.

2. *Thanks for pointing out this additional non-local effect of eddies on chlorophyll and the associated references. To accommodate your comment and the main comment of the second reviewer, Volker Strass, we have included in the Discussion section a paragraph on the potential effects of eddies on chlorophyll/biogeochemical rates which we do not consider in our analysis (see below); further, we have added the attribute "local" to "stirring" in several places throughout the manuscript; and, yes, indeed, we tend to find positive anomalies, both for cyclones and anticyclones downstream of shelves (see also below, p19L7ff).*

3. *We added in the Discussion section*
   *p19L7ff: "A possible explanation next to advection of Chl is the offshore advection of iron trapped in the nearshore region by eddies that fuels extra growth in the offshore waters, as suggested e.g., for Haida eddies in the North Pacific [Xiu et al., 2011], or for eddies passing the Kerguelen Plateau [D'Ovidio et al., 2015]. A substantial effect of trapping to cause $\delta Chl$ of eddies originating from boundary currents corroborates previous results [Gaube et al., 2014].", and*
   *p20L22: "Further, we may underestimate the overall effect of eddies on Chl also because of additional effects of eddies that are not considered in our analysis. Such effects include the impact of smaller mesoscale features, and of submesoscale processes near the edges of eddies [Woods, 1988, Strass, 1992, Martin et al., 2002, Lévy, 2003, Klein and Lapeyre, 2009, Siegel et al., 2011], e.g., due to eddy-jet interactions and associated horizontal shear-induced patches of up- and downwelling. Such features are included in our analysis only insofar they have rectified effects on the larger mesoscale Chl patterns resolved by the data we use. Another effect we do not consider is non-local stirring [D'Ovidio et al., 2015], the contribution of eddies to lateral dispersion outside the eddies' cores in interaction with the ambient flow. This effect, for instance, shapes iron plumes downstream of shelves along the ACC, thus preconditioning Chl blooms [Ardyna et al., 2017]. Therefore, we note that the overall effect of eddies on biogeochemical rates may be larger than suggested by our analysis of the mesoscale, local imprint of eddies on Chl.";*
   *further we added the reference of Ardyna et al in the context of the non-zonality of the Chl p9L7ff: "A few exceptions break this mostly zonal picture for Chl [Ardyna et al., 2017], and also for $\delta Chl$.", and of the seasonality of the imprint of eddies, p10L6.*

**Minor comments**

**Comment b**

1. timescale of chlorophyll: chlorophyll is just a pigment. Referring to the timescale of a bloom, or of phytoplankton demography, should be more appropriate.

2. *Thanks for the comment, we have adjusted the text accordingly (see below).*

3. *We have changed time scale of Chl to time scale of phytoplankton demography in the section* Causes of $\delta$Chl by advective processes *(p3L6ff).*

**References**

M. Ardyna, H. Claustre, J. B. Sallée, F. D'Ovidio, B. Gentili, G. van Dijken, F. D'Ortenzio, and K. R. Arrigo. Delineating environmental control of phytoplankton biomass and phenology in the Southern Ocean. *Geophys. Res. Lett.*, 44(10):5016–5024, 2017. ISSN 19448007. doi: 10.1002/2016GL072428.

F. D'Ovidio, A. Della Penna, T. W. Trull, F. Nencioli, M. I. Pujol, M. H. Rio, Y. H. Park, C. Cotté, M. Zhou, and S. Blain. The biogeochemical structuring role of horizontal stirring: Lagrangian perspectives on iron delivery downstream of the Kerguelen Plateau. *Biogeosciences*, 12(19):5567–5581, 2015. doi: 10.5194/bg-12-5567-2015.

P. Gaube, D. McGillicuddy, D. Chelton, M. J. Behrenfeld, and P. Strutton. Regional variations in the influence of mesoscale eddies on near-surface chlorophyll. *Journal of Geophysical Research: Oceans*, 119:8195–8220, 2014. ISSN 0028-0836. doi: 10.1002/2014JC010111.

P. Klein and G. Lapeyre. The oceanic vertical pump induced by mesoscale and submesoscale turbulence. *Annu. Rev. Mar. Sci.*, 1:351–375, 2009. doi: 10.1146/annurev.marine.010908.163704.

M. Lévy. Mesoscale variability of phytoplankton and of new production: Impact of the large-scale nutrient distribution. *J. Geophys. Res.*, 108(C11):3358, 2003. ISSN 0148-0227. doi: 10.1029/2002JC001577.

A. P. Martin, K. J. Richards, A. Bracco, and A. Provenzale. Patchy productivity in the open ocean. *Global Biogeochem. Cycles*, 16(2):1025, 2002. doi: 10.1029/2001GB001449.

D. A. Siegel, P. Peterson, D. J. McGillicuddy, S. Maritorena, and N. B. Nelson. Bio-optical footprints created by mesoscale eddies in the Sargasso Sea. *Geophysical Research Letters*, 38:L13608, 2011. ISSN 0094-8276. doi: 10.1029/2011GL047660.

V. H. Strass. Chlorophyll patchiness caused by mesoscale upwelling at fronts. *Deep Sea Res. Part A. Oceanogr. Res. Pap.*, 39(1):75–96, 1992. doi: 10.1016/0198-0149(92)90021-K.

J. Woods. Scale upwelling and primary production. In *Towar. a Theory Biol. Interact. World Ocean*, pages 7–38. Springer Netherlands, Dordrecht, 1988. doi: 10.1007/978-94-009-3023-0$_2$.

P. Xiu, A. P. Palacz, F. Chai, E. G. Roy, and M. L. Wells. Iron flux induced by Haida eddies in the Gulf of Alaska. *Geophysical Research Letters*, 38(13): L13607, 2011. ISSN 0094-8276. doi: 10.1029/2011GL047946.

---

## Author Comment (AC2) · 18 May 2018

Reviewer Comment 2, V. Strass, and *author response*:

**Major comments**

**General Appraisal**

**Comment a**

1. The paper presents the results of a truly impressive data analysis of the effects of mesoscale eddies on sea surface chlorophyll in the Southern Ocean, comprising an extensive and widely new look into the regional and seasonal variation of these effects. In respect of how those interesting results are set into scientific context, however, the paper has severe weaknesses. I think these weaknesses can be overcome by rewriting major parts of the manuscript, sections 1, 4 and 5 in particular.

2. *Thank you for the positive assessment regarding our analysis, and for providing comments and references to better set our paper into context.*

3. *We have included the additional references and modified sections 1, 4 and 5 according to the above comment and the detailed comments below.*

**Specific Comments**

**Comment b**

1. The mesoscale ocean dynamics govern the range from a few kilometres to a few hundreds of kilometres horizontally. The data used in the study by Frenger et al., collected by satellite remote sensing, provide a horizontal resolution of 1/3 of a degree for eddies (Aviso SLA) and $0.25°$ for the concentration of sea surface chlorophyll (ESA GlobColour Project product), i.e. approx. 37 km and 28 km in latitude, respectively. In consequence, only the larger fraction of mesoscale eddies is investigated. This needs, but is not yet, be made clear in the paper.

2. *We have included a sentence in the Method and Discussion section to clarify the constraints due to the resolution of the satellite data.*

3. *p6L10 "The resolution capacity of Aviso SLA allows to analyze the larger mesoscale eddies [Chelton et al., 2011]."*

**Comment c**

1. Eddies, or the mesoscale dynamics in general, affect phytoplankton hence the chlorophyll concentration in various ways, particularly by time-variable horizontal and vertical advection and associated transports of nutrients,

and by vertical current shears that control stratification and subduction hence the light environment which phytoplankton cells experience. (In the Southern Ocean, where most macro-nutrients are abundant, it is likely the mesoscale upwelling of the primary production-limiting micro-nutrient iron that enhances biological production in the ACC with its meandering fronts; Hense, et al., Regional ecosystem dynamics in the ACC: Simulations with a three-dimensional ocean-plankton model , J. Mar. Systems, 2003.) Vertical velocities, and therefore possible upwelling of nutrients, but are known be most intense at the smallest part ($\leq$10 km) of the mesoscale range (Martin et al., Patchy productivity in the open ocean, Global Biogeochem. Cycles 2002; Lévy, Mesoscale variability of phytoplankton and of new production: Impact of the large scale nutrient distribution, J. Geophys. Res., 2003; Klein & Lapeyre, The oceanic vertical pump induced by mesoscale and submesoscale turbulence, Annu. Rev. Mar. Sci. 2009). The relevance of those small scales has been noted initially by Woods (Mesoscale upwelling and primary production, in Toward a theory on biological–physical interactions in the world ocean, ed. B. J. Rothschild, Dordrecht Kluwer,1988), who raised the hypothesis that key to understanding the plankton patchiness which was revealed with the advent of satellite chlorophyll images, lies in the dynamics of mesoscale jets, where dynamical constraints limit upwelling to horizontal dimensions of about ten kilometres. This hypothesis received first observational support in 1992 (Strass, Chlorophyll patchiness caused by mesoscale upwelling at fronts, Deep-Sea Res. I). These latter two publications, by the way, would close the glaring time gap of the literature review given in the Introduction (p.1, lines 20 – 22.) between the cited advent of satellite chlorophyll images (Gower et al. 1980) and Doney (2003).

2. *Thank you for these points and the additional references. Our objective is not to discuss in our paper processes $\leq$10 km, that is submesoscale processes (which appear to be the focus of the points above/below). These are, even though connected to the mesoscale, a separate field of study, and we certainly do not wish to claim to resolve direct submesoscale effects in the data we use. Hence, we have included the suggested references and tried to make very clear throughout the text that we focus on the larger mesoscale as resolved by the satellite data we use for the study. Further, we have included in the Discussion section a paragraph on the potential effects of eddies on chlorophyll/biogeochemical rates which we do not consider in our analysis (see also Reviewer Comment 1 by Francesco d'Ovidio, and below)*

3. *p20L22: "Further, we may underestimate the overall effect of eddies on Chl also because of additional effects of eddies that are not considered in our analysis. Such effects include the impact of smaller mesoscale features, and of submesoscale processes near the edges of eddies [Woods, 1988, Strass, 1992, Martin et al., 2002, Lévy, 2003, Klein and Lapeyre, 2009, Siegel et al., 2011], e.g., due to eddy-jet interactions and associated horizontal shear-induced patches of up- and downwelling. Such features are*

*included in our analysis only insofar they have rectified effects on the larger mesoscale Chl patterns resolved by the data we use. Another effect we do not consider is non-local stirring [D'Ovidio et al., 2015], the contribution of eddies to lateral dispersion outside the eddies' cores in interaction with the ambient flow. This effect, for instance, shapes iron plumes downstream of shelves along the ACC, thus preconditioning Chl blooms [Ardyna et al., 2017]. Therefore, we note that the overall effect of eddies on biogeochemical rates may be larger than suggested by our analysis of the mesoscale, local imprint of eddies on Chl."*.

**Comment d**

1. The presumably most important horizontal scale for stimulating phytoplankton growth, as explained above, unfortunately is not resolved by the present study. Moreover, most of the above-mentioned studies have demonstrated that up- and downwelling predominately are driven by changes in time of the mesoscale flow field (related to the development of frontal meanders due to baroclinic instability, frontogenesis by eddyeddy interaction etc.). For the ACC, Strass et al. (Mesoscale frontal dynamics: Shaping the environment of primary production in the Antarctic Circumpolar Current, Deep-Sea Research II, 2002) have shown with an in situ study that the acceleration/deceleration of a frontal jet by interaction with an eddy creates a pattern of up- and downwelling cells and of chlorophyll patches on a much smaller horizontal scale than that of the involved eddy. Frenger et al. in their present study, however, analysed only eddies that were tracked over at least three weeks, hence eddies which were not subject to much change in time. Both by the selection of eddies of larger size and of low temporal change, Frenger and co-authors likely introduce a bias towards eddies of rather limited impact on biological production and biogeochemical rates. Their conclusion that eddydriven stirring and trapping dominate over biogeochemical effects therefore seems not robust but rather a result of the horizontal/time scale bias. This requires an honest and thorough discussion.

2. *See also response above. We agree that additional effects of the smaller mesoscale and of the submesoscale on shorter time scales are important for biogeochemical rates; even with our focus on the larger mesoscale, we were starting out with the hypothesis that these eddies affect biogeochemical rates. Yet, our conclusion is valid, that the Chl imprint we find of the scales that our data do resolve can be explained largely by advection. This conclusion does not negate that effects on biogeochemical rates may be at play, too. To try to accommodate your concern, and the main comment by Francesco d'Ovidio, that overall biogeochemical effects may be underestimated, we included a paragraph in the Discussion section (see response to comment above), and highlighted throughout the text that we focus on the larger mesoscale.*

**Comment e**

1. Throughout their ms Frenger and co-authors associate cyclonic eddies with thermocline lifting and anticyclonic eddies with thermocline deepening. Undisputable is that cyclones display a lifted thermocline and anticyclones a deepened thermocline. However, whether or not the thermocline moves up or down after eddies have been fully formed is in contestation. It may well be the reverse of the indicated way, i.e. that during eddy slow-down due to processes such as eddy-induced Ekman pumping, the thermocline in cyclones moves downward and in anticyclones upward (e.g. Gaube et al., 2014). I therefore recommend the authors use to a more careful wording, i.e. lifted/deepened instead of lifting/deepening.

2. *We much appreciate this comment.*

3. *We replaced "lifting/deepening" with "lifted/deepened" throughout the text.*

**Comment f**

1. On p. 20, lines 30-31 the authors bring forward the argument that anticyclones cause an abatement of grazing pressure, without providing a reference. In general I would doubt that a reference for this argument exists, which could be considered representing the widely accepted and unquestioned state of knowledge regarding mesoscale variability of grazing. Therefore, I consider this argument pure speculation, and suggest remove it from the ms.

2. *We would like to keep this hypothesis, yet, to make clear that it is a mere hypothesis/speculation, we rephrase the respective sentences mentioning grazing (see below).*

3. *p19L35: "Hypothetically, an alleviation of grazing pressure [...]", and p21L18: "and, more speculatively, with an abatement of grazing pressure caused by anticyclones via deepened mixed layers,".*

**Technical Issues**

**Comment g**

1. Fig. 4 should be enlarged to full-page size to enhance its readability in the pdf-version of the paper, and the caption therefore be shifted to next page, if possible.

    *2. We agree; yet, we do not see how the Biogeosciences LaTeX template (which does not allow to use additional packages) would allow us to do this; we will ask the editorial/production team at the publication stage to enlarge the figure.*

**Comment h**

1. Caption Fig 6 associates autumn with the months January to May, what is certainly not correct. If the given months are valid, then the season should be termed high summer – autumn or so.

2. *Thank you.*

3. *We have adjusted the caption of Fig 6 to "**a** high summer to autumn ACC waters R2 (SSH -60 to -40 cm, January to May) and **b** R3 northern ACC winter to early spring deep mixed layer waters (SSH -50 to -15 cm, July to September);"*

**References**

M. Ardyna, H. Claustre, J. B. Sallée, F. D'Ovidio, B. Gentili, G. van Dijken, F. D'Ortenzio, and K. R. Arrigo. Delineating environmental control of phytoplankton biomass and phenology in the Southern Ocean. *Geophys. Res. Lett.*, 44(10):5016–5024, 2017. ISSN 19448007. doi: 10.1002/2016GL072428.

D. B. Chelton, M. G. Schlax, and R. M. Samelson. Global observations of nonlinear mesoscale eddies. *Progress in Oceanography*, 91:167–216, oct 2011. ISSN 00796611. doi: 10.1016/j.pocean.2011.01.002.

F. D'Ovidio, A. Della Penna, T. W. Trull, F. Nencioli, M. I. Pujol, M. H. Rio, Y. H. Park, C. Cotté, M. Zhou, and S. Blain. The biogeochemical structuring role of horizontal stirring: Lagrangian perspectives on iron delivery downstream of the Kerguelen Plateau. *Biogeosciences*, 12(19):5567–5581, 2015. doi: 10.5194/bg-12-5567-2015.

P. Klein and G. Lapeyre. The oceanic vertical pump induced by mesoscale and submesoscale turbulence. *Annu. Rev. Mar. Sci.*, 1:351–375, 2009. doi: 10.1146/annurev.marine.010908.163704.

M. Lévy. Mesoscale variability of phytoplankton and of new production: Impact of the large-scale nutrient distribution. *J. Geophys. Res.*, 108(C11):3358, 2003. ISSN 0148-0227. doi: 10.1029/2002JC001577.

A. P. Martin, K. J. Richards, A. Bracco, and A. Provenzale. Patchy productivity in the open ocean. *Global Biogeochem. Cycles*, 16(2):1025, 2002. doi: 10.1029/2001GB001449.

D. A. Siegel, P. Peterson, D. J. McGillicuddy, S. Maritorena, and N. B. Nelson. Bio-optical footprints created by mesoscale eddies in the Sargasso Sea. *Geophysical Research Letters*, 38:L13608, 2011. ISSN 0094-8276. doi: 10.1029/2011GL047660.

V. H. Strass. Chlorophyll patchiness caused by mesoscale upwelling at fronts. *Deep Sea Res. Part A. Oceanogr. Res. Pap.*, 39(1):75–96, 1992. doi: 10.1016/0198-0149(92)90021-K.

J. Woods. Scale upwelling and primary production. In *Towar. a Theory Biol. Interact. World Ocean*, pages 7–38. Springer Netherlands, Dordrecht, 1988. doi: 10.1007/978-94-009-3023-0$_2$.

---

## Author Response (AR1)

Reviewer Comment 1, F. d'Ovidio, and *author response*:

**Major comments**

The paper provides an analysis of the role of mesoscale eddies in the Southern Ocean on primary production, in terms of chlorophyll anomalies detected by remote sensing associated to them. Methodologically, the paper follows very closely some previous works, in particular Gaube et al. 2014, which were more focused on the global ocean. In respect to previous works, this manuscript has fine tuned the methodology, and discussed the results in the specific context of the Southern Ocean. The main original result in this manuscript is the finding of a strong seasonal signal in the mesoscale imprint on chlorophyll anomalies. This result and other more incremental findings are not surprising, but are very well discussed in terms of the previous literature and in terms of the biogeochemical activity of the Southern ocean (with possibly one direction of improvement which is described below). As a consequence, I find this manuscript as a useful contribution to the understanding of the role of mesoscale eddies on primary production in the Southern Ocean, even in its current form.

*Thank you for the supportive review. We have amended the manuscript accoridng to the detailed comments below.*
*Additionally we have added minor changes to the text aimed at further clarifying the manuscript that were not in response to particular reviewer comments. These changes did not alter any qualitative or quantitative conclusions from the original manuscript.*

**Comment a**

1. There is however one issue that may improve further the manuscript. One of the main concept treated by the paper, is stirring, and in particular the imprint of stirring induced by the mesoscale eddies on the mesoscale anomalies of the chlorophyll field. The manuscript explains that the stirring created by a mesoscale eddy can create a local deformation of a pre-existing chlorophyll gradient and I agree with this statement. However, this is not all about stirring. In fact, if I think to the imprint of stirring and chlorophyll in the Southern ocean, the main effect that comes to my mind is not the generation of local chlorophyll anomalies, but the huge plumes of chlorophyll induced when stirring by mesoscale eddies modulates iron delivery in a non-local way, preconditioning the blooms of this region. An analysis of this effect is not in the scope of this paper, and it has been done elsewhere (for instance, d'Ovidio et al. Biogeosciences 12, 2015; Ardyna et al. GRL 44, 2017). Nevertheless, I feel that the submitted manuscript should stress more that what the authors intend here for eddy stirring, is only the local effect of stirring, while other non-local effects of stirring by mesoscale activity also exist, and actually they are a prominent driver of the bloom extension and intensity in the Southern Ocean. In fact, it would

be interested to know whether there is a signature of this non-local stirring effect in the analysis presented, for instance by finding stronger anomalies downstream of likely iron sources like the continental shelves present in the region. Or as a possible alternative explanation of the asymmetries in the chlorophyll anomalies. I am certainly biased in this comment by my own work on the subject, therefore the authors are free to find some other papers instead of the two indicated above to add to their discussion. But in any case, I feel that the discussion on stirring merits to be extended.

2. *Thanks for pointing out this additional non-local effect of eddies on chlorophyll and the associated references. To accommodate your comment and the main comment of the second reviewer, Volker Strass, we have included in the Discussion section a paragraph on the potential effects of eddies on chlorophyll/biogeochemical rates which we do not consider in our analysis (see below); further, we have added the attribute "local" to "stirring" in several places throughout the manuscript; and, yes, indeed, we tend to find positive anomalies, both for cyclones and anticyclones downstream of shelves (see also below, p19L9ff).*

3. *We added in the Discussion section*
   *p19L16ff: "An additional possible explanation is the offshore advection of iron trapped in the nearshore region by eddies that fuels extra growth in the offshore waters, as suggested e.g., for Haida eddies in the North Pacific [Xiu et al., 2011], or for eddies passing the Kerguelen Plateau [D'Ovidio et al., 2015]."*
   *p20L25: "Furthermore, we may underestimate the overall effect of mesoscale eddies on Chl also because of additional effects of mesoscale eddies that are not considered in our analysis. Such effects include the impact of smaller mesoscale features, and of submesoscale processes near the edges of eddies [Woods, 1988, Strass, 1992, Martin et al., 2002, Lévy, 2003, Klein and Lapeyre, 2009, Siegel et al., 2011], e.g., eddy-jet interactions and associated horizontal shear-induced patches of up- and downwelling. Such features are included in our analysis only insofar as they have rectified effects on the larger mesoscale Chl patterns resolved by the data we use. Another effect we do not consider is non-local stirring [D'Ovidio et al., 2015], the contribution of eddies to lateral dispersion outside the eddies' cores in interaction with the ambient flow. This effect, for instance, shapes iron plumes downstream of shelves along the ACC, thus preconditioning Chl blooms [Ardyna et al., 2017]. Therefore, we note that the overall effect of mesoscale eddies on biogeochemical rates may be larger than suggested by our analysis of the mesoscale, local imprint of eddies on Chl. Finally, we note that our analysis does not include the effect of submesoscale processes outside eddies as well as any unstructured turbulence in general.". Further we added the reference of Ardyna et al in the context of the non-zonality of the Chl p10L1ff: "A few exceptions break this mostly zonal picture for Chl [Ardyna et al., 2017], and also for $\delta$Chl.", and of the seasonality of the imprint of eddies, p10L10.*

**Minor comments**

**Comment b**

1. timescale of chlorophyll: chlorophyll is just a pigment. Referring to the timescale of a bloom, or of phytoplankton demography, should be more appropriate.

2. *Thanks for the comment, we have adjusted the text accordingly (see below).*

3. *We have changed time scale of Chl to time scale of phytoplankton biomass changes in the section* Causes of δChl by advective processes *(p3L13ff).*

[revised manuscript text omitted]

---

## Author Response (AR2)

**Editor comment:**

Dear Dr Frenger,

I am happy to inform you that the revised version of your manuscript is accepted for publication in Biogeosciences. In the annotated document please find some minor suggestions that might benefit the overall presentation of the manuscript.

Thank you for supporting Open Access journals like Biogeosciences.

Christine Klaas

**Author response:**

Dear Dr Klaas,

We much appreciate that our manuscript has been accepted for publication in Biogeosciences.

Thank you for the careful read of our manuscript and suggestions to further clarify the manuscript. We have adjusted the manuscript accordingly. Further, we have removed a few more typos and unclear phrasings, all of which do not affect the content of the manuscript. Finally, we have added an additional sentence in the Method Section to further simplify a potential reproduction of our Method (p7L12):

"We then assign a sign to $\hat{\delta} Chl_{stir}$ according to the sign of the meridional Chl gradient and the cyclonicity of the eddy, given the intrinsic westward propagation of eddies: We anticipate that, e.g., a southern hemispheric counterclockwise-rotating, i.e. anticyclonic eddy under conditions of northward increasing Chl will be associated with positive $\delta Chl$ in its core (Fig. 1a, left column). In contrast, under the same ambient Chl conditions we anticipate negative $\delta Chl$ for cyclones."

Best regards,
Ivy Frenger